# Study of the Interactions between Muscle Fatty Acid Composition, Meat Quality-Related Genes and the Ileum Microbiota in Tibetan Sheep at Different Ages

**DOI:** 10.3390/foods13050679

**Published:** 2024-02-23

**Authors:** Fanxiong Wang, Yuzhu Sha, Xiu Liu, Yanyu He, Jiang Hu, Jiqing Wang, Shaobin Li, Pengyang Shao, Xiaowei Chen, Wenxin Yang, Qianling Chen, Min Gao, Wei Huang

**Affiliations:** 1Gansu Key Laboratory of Herbivorous Animal Biotechnology, College of Animal Science and Technology, Gansu Agricultural University, Lanzhou 730070, China; m19893318751@163.com (F.W.); shayz@st.gsau.edu.cn (Y.S.); huj@gsau.edu.cn (J.H.); wangjq@gsau.edu.cn (J.W.); lisb@gsau.edu.cn (S.L.); shaopengyang666@163.com (P.S.); cxw20002022@163.com (X.C.); aaaaa0108@163.com (W.Y.); chenqianling223@163.com (Q.C.); gm12017101@163.com (M.G.); 18294737108@163.com (W.H.); 2School of Fundamental Sciences, Massey University, Palmerston North 4410, New Zealand; y.h@massey.ac.nz

**Keywords:** Tibetan sheep, muscle fatty acids, ileum microbiota, meat quality-related genes, gene expression

## Abstract

The intestinal microbiota of ruminants is an important factor affecting animal production and health. Research on the association mechanism between the intestinal microbiota and meat quality of ruminants will play a positive role in understanding the formation mechanism of meat quality in ruminants and improving production efficiency. In this study, the fatty acid composition and content, expression of related genes, and structural characteristics of the ileum microbiota of ewes of Tibetan sheep at different ages (4 months, 1.5 years, 3.5 years, and 6 years) were detected and analyzed. The results revealed significant differences in fatty acid composition and content in the muscle of Tibetan sheep at different ages (*p* < 0.05); in addition, the content of MUFAs in the longissimus dorsi muscle and leg muscle was higher. Similarly, the expressions of muscle-related genes differed among the different age groups, and the expression of the *LPL*, *SCD*, and *FABP4* genes was higher in the 1.5-year-old group. The ileum microbiota diversity was higher in the 1.5-year-old group, the *Romboutsia* abundance ratio was significantly higher in the 1.5-year-old group (*p* < 0.05), and there was a significant positive correlation with oleic acid (C18:1n9c) (*p* < 0.05). In conclusion, the content of beneficial fatty acids in the longissimus dorsi muscle and leg muscle of Tibetan sheep was higher at 1.5 years of age, and the best slaughter age was 1.5 years. This study provides a reference for in-depth research on the mechanism of the influence of the gut microbiota on meat quality and related regulation.

## 1. Introduction

Tibetan sheep is one of China’s three original sheep breeds (Mongolian sheep, Tibetan sheep, and Kazakh sheep); it is mainly bred in the Tibetan Plateau region, which features high-altitude, low-temperature, low-oxygen, and high-ultraviolet-radiation conditions, and has one of the largest stocks of sheep breeds in China [1] (estimated at 25 million). The special highland environment has resulted in the meat of Tibetan sheep being low in cholesterol, low in fat, and rich in nutrients, making it popular among consumers [2]. With the increasing demand for mutton, there has also been a demand for higher quality mutton [3], and with the development of production intensification, improving the growth performance and carcass quality of Tibetan sheep can improve economic efficiency [4]. Fatty acids are important chemicals that make up fat and are very important aromatic substances that are important indicators of lamb quality [5]. The structure and composition of fatty acids not only influence muscle fat formation but also have a very important effect on the flavor, tenderness, and juiciness of meat [6]. Unsaturated fatty acids (UFAs) not only have a significant impact on the production of flavor substances and the nutritional status of meat but also have specific functions that are important in combatting cancer, reducing fat, preventing cardiovascular disease, and preserving human health [7,8]; however, UFAs are prone to chemical reactions with oxygen in the air, resulting in the production of compounds that adversely affect the quality of meat. The products of lipid oxidation are associated with a loss of odor and color and may ultimately affect the safety of meat [9].

The quality of lamb meat is affected not only by breed, sex, feeding management, and environmental factors but also by age [10,11,12]. Age has a significant effect on fat and fatty acid changes, with fat deposition in lambs being significantly lower than that in adult sheep during early growth and development and flavor development generally occurring in adulthood. Age affects marbling, muscle fiber density, diameter, shear strength, and water loss in lamb [13]. Age is also an important factor in driving the maturation of gut microbes [14,15]. Changes in pig gut microbiota over time have been found, with nursery pigs having the lowest gut microbiota diversity compared to older pigs [16]. The gut microbiota is involved in biohydrogenation and isomerization in vivo and may ultimately influence muscle fatty acid deposition by altering the location of fatty acids in the digesta [17]. It was found that ruminants have very low levels of PUFAs due to biohydrogenation by the gut microbiota. Microbiota metabolites are also involved in the composition of meat flavor components, especially in ruminants, which are directly or indirectly related to microbiota fermentation products and microbiota metabolism [18]. Interactions between the host and the gut microbiota affect metabolite deposition in muscle [19]. More efficient animals identified in previous studies may produce more short-chain fatty acids and microbiota proteins by fermentation of highly fermentable carbohydrates in the rumen [20], and upon reaching a portion of the small intestine, undigested nutrients promote the growth of small intestine microbes, increasing the efficiency of production. Studies in cows have shown that a less diverse gut microbiota is associated with the lower efficiency of milk, milk fat, and milk protein production [21]. When nutrients reach the small intestine, microorganisms use these nutrients to release more energy, which increases the efficiency of production. The metabolic mechanisms of muscle are complex, and host genes are also involved in the regulation of meat and fatty acid metabolism. For example, lipoprotein esterase (*LPL*) is the rate-limiting enzyme of triacylglycerol metabolism and is considered a functional candidate gene for the regulation of fatty acid composition [22,23]. Stearoyl coenzyme A desaturase (*SCD*) plays an important role in lipid metabolism [24]. The regulation of its expression is important for the maintenance of the normal physiological state of animals and the stability of the lipid internal environment in vivo [25]. *FABP* has an affinity for long-chain fatty acids, so it can preferentially bind and transport long-chain fatty acids (LCFAs), thus promoting lipidation and triglyceride (TG) synthesis [26]. *FASN* is the gene encoding a fatty acid synthase that controls the ab initio biosynthesis of LCFAs and catalyzes the synthesis of long-chain SFAs, and elevated *FASN* expression significantly increases TG deposition in vivo [27]. Therefore, controlling muscle development, fat deposition, and fatty acid composition is important for the production of meat products. However, studies on the interactions between intestinal microbes and muscle fatty acids and related gene expression in Tibetan sheep of different ages are rare. Therefore, in this study, the characteristics of muscle fatty acids, the expression of muscle fatty acid-related genes, and their interactions with the ileum microbiota of Tibetan sheep at different ages were investigated to determine the effects of age on the ileum microbiota, host genes, muscle fat deposition, fatty acid composition and content of Tibetan sheep. This study provides a basis for assessing the nutritional value, industrial development, and utilization of Tibetan sheep meat.

## 2. Materials and Methods

### 2.1. Test Animals and Sample Collection

The study was conducted on grazing Tibetan sheep in Haiyan County, Hai bei Prefecture, Qinghai Province (Altitude 3500 m), all sheep from the same herd (about 300). Tibetan sheep with good health were randomly selected at 4 months old (n = 6), 1.5 years old (n = 6), 3.5 years old (n = 6) and 6 years old (n = 6). All sheep were grazed on the same pasture, under the traditional natural grazing management of the area, without any supplementary feeding. The samples were collected in August 2020, and the forage types and nutrient levels in the pasture were detailed in the team’s previous research (Table 1). In accordance with the ethical approval requirements of the Ethics Committee and local traditional slaughtering and sampling methods, after the four age groups of Tibetan sheep were slaughtered by neck bleeding respectively, the gastrointestinal organs were immediately removed, and the ileum contents were collected, about 50 mL of the ileum contents were collected from each sheep, which were divided into cryopreservation tubes and quickly placed into liquid nitrogen tanks for freezing, and brought back to the laboratory at −80 °C for preservation and used for the subsequent 16S rRNA analyses. Then, samples of longissimus dorsi muscle (the muscle between the penultimate and second thoracic vertebrae), foreleg muscle (triceps humerus), and hindleg muscle (biceps femoris) were taken and divided into freezer-storage tubes, quickly put into a liquid nitrogen tank and brought back to the laboratory for storage at −80 °C for subsequent RNA extraction. Then 500 g samples of longissimus dorsi muscle, foreleg muscle, and hindleg muscle were collected and brought back to the laboratory in an ice box at −20 °C for the subsequent determination of fatty acids.

### 2.2. Extraction and Determination of Fatty Acids

The meat is thawed at room temperature, the skin is removed, and the fat on the surface is removed with a knife. Each sample is placed in a mortar and ground with liquid nitrogen. Then 1.0 g powder sample was taken and weighed in 10 mL plug tube. 0.7 mL KOH solution with a concentration of 10 mol L^−1^ and 5.3 mL methanol with a concentration of anhydrous [for methanol analysis (chromatography)] were added, respectively, washed at 55 °C for 1.5 h, and the test tube was shaken every 20 s for 5 s. The tube was removed from the water bath, cooled to below room temperature under running water, and added 0.58 mL H_2_SO_4_ solution with a concentration of 12 mol L^−1^. The free fatty acid methyl ester was treated in a constant temperature water bath at 55 °C for 1.5 h and shaken every 20 min for 5 s. After the water bath is heated, removed the tube, cooled it with tap water to below room temperature, added 3 mL *n*-hexane, shook well, transferred the mixture to the centrifuge tube. The supernatant was filtered into the sample bottle with an organic phase filtration membrane (0.45 μm) and then subjected to gas chromatography (GC-2010 +; Shimadzu Corporation, Kyoto, Japan) test. The chromatographic conditions refer to Wu’s method [29].

The fatty acid composition and content of muscle fatty acids were examined by gas chromatograph (GC), and three parallel samples of the same sample were tested. The average value was taken as the result of the test for that sample. The fatty acids were determined according to the relative retention time of the fatty acid methyl ester standards, and the relative percentage content of each fatty acid was determined using peak area normalization. The relative percentages of saturated fatty acid (SFA), unsaturated fatty acid (UFA), monounsaturated fatty acid (MUFA), polyunsaturated fatty acid (Polyunsaturated fatty acid (PUFA), and the ratio of PUFA to SFA (P/S).

### 2.3. RNA Extraction and Detection

Total RNA was extracted from longissimus dorsi muscle and leg muscle of Tibetan sheep with TRIzol reagent. Before the extraction of total RNA, the extraction consumables were autoclaved, and the consumables such as sterile enzyme-free gun tip and centrifugal tube were used. The frozen tissue samples were extracted from the frozen storage tube; about 100 mg of the tissue samples were clipped and put into a mortar (adding new liquid nitrogen continuously), ground into powder (no obvious particles), and transferred to the centrifugal tube. Then, 1000 µL *AG RNAex Pro* Reagent was added and let stand for 5 min (blow until the lysate is transparent); the centrifuge tube was covered and placed in a centrifuge (12,000× *g*). After that, the follow-up test operation was carried out according to the kit guide, and the test process was carried out on the ultra-clean workbench. After the extraction, the concentration and purity of 2 µL RNA samples were detected by ultramicro spectrophotometer (Thermo Nano drop-2000), and the integrity of RNA was detected by agarose gel electrophoresis. RNA concentration (ng/µL) = OD260 × dilution factor × 40. Reverse transcription kits (HiScript^®^ II Q RT SuperMix for qPCR; The cDNA was synthesized in Nanjing, China, and the detailed operation was carried out in strict accordance with the kit instructions. After cDNA synthesis, Primer5.0 software was used to design gene primers (*β-actin* as internal reference gene), and the primer information is shown in Table 2. The relative quantification of the relevant genes was performed using a real-time fluorescence quantitative PCR instrument. Reaction conditions: pre-denaturation at 95 °C for 30 s; cycling reaction at 95 °C for 10 s and 60 °C for 30 s for 40 cycles; solubilization curves (95 °C for 15 s, 60 °C for 60 s and 95 °C for 15 s). Reaction system: 20 µL system containing 2× Cham Q Universal SYBR qPCR Master Mix, cDNA template and upstream and downstream primers. *β-actin* was used as the internal reference gene for correction, and the data were analyzed by the method of 2^−∆∆CT^.

### 2.4. Biostatistical Analysis

The ileum microbiota DNA was extracted using the DNA extraction kit MN NucleoSpin 96 Soi (Omega, Shanghai, China). A total of 250–500 mg of fresh sample was transferred to a 2.0 mL inlet centrifuge tube (with grinding beads added), and 700 µL of SL2 was added (SL2 is preferred for the first extraction, and if results are poor, SL1 can be tried for re-extraction); and then 150 µL of Enhancer SX, centrifuge tube was covered tightly and vortexed to mix. Afterward, the kit extraction steps were followed: lysis of the sample, precipitation to remove impurities, filtration to remove inhibitors, and to the final DNA elution. Concentration and purity were measured using an ultra-micro spectrophotometer (Thermo Nano Drop-2000) and detected by agarose gel electrophoresis. The DNA samples were stored at −80 °C. The ileum microbiota characteristics were obtained by PCR amplification of the V3-V4 region of the 16S rRNA gene. The universal primers were 338F (5’-ACTCCTACGGGGAGGCAGCAG-3’) and 806R (5’-GGACTACH VGGGTWTCTAAT-3’), and the PCR amplification conditions were as follows: 95 °C pre-denaturation for 5 min; 95 °C denaturation for 30 s, 55 °C annealing for 30 s, 72 °C extension for 40 s, 25 cycles, 72 °C extension for 7 min. The PCR products from the previous step were mixed according to the electrophoretic quantification results in accordance with the mass ratio of 1:1. After mixing, the samples were purified by OMEGA DNA purification columns, and then the target fragments were cut and recovered by 120 V 40 min electrophoresis on a 1.8% agarose gel. Up-sequencing: library building sequencing analysis was performed on Illumina MiSeq (Illumina, San Diego, CA, USA) platform. Raw sequencing data were assessed for quality, and raw sequencing reads were denoised, double-ended spliced (FLASH, version 1.2.11), quality filtered (Trimmomatic, version 0.33), and chimeras removed (UCHIME, version 8.1). High-quality effective data (Effective Tags) were clustered with the 2013 Greengenes (version 13.8) ribosome database at the 97% similarity level using Usearch software (version 10.0), and OTUs were filtered using 0.005% of the number of all sequences sequenced as a threshold, and based on the OTUs were filtered by using 0.005% of all sequenced sequences as a threshold, and species annotation and taxonomic analysis of OTUs were performed based on the Silva (Bacteria 16S) database. Alpha diversity analysis of OTUs was performed by Mothur (version V.1.30), and the Rarefaction Curve, Shannon Index and cumulative relative abundance of species were plotted. Beta diversity was analyzed by QIIME, and Principal coordinates analysis (PCoA), NonMetricMulti-Dimensional Scaling, (NMDS), and Anosim (analysis of similarities) were analyzed to compare the degree of similarity in species diversity across samples. Biomarker of intergroup differences was obtained by LEfSe analysis of evolutionary meristems. Finally, PICRUSt2 software was used to analyze the functional differences between subgroups by comparing the species composition information obtained from the 16S sequencing data; differences and changes in metabolic pathways of functional genes of microbial communities between subgroups of samples were observed by KEGG (Kyoto Encyclopedia of Genes and Genomes) difference analyses. Predicting differences and changes in protein function in prokaryotes between different subgroups by COG (Clusters of Orthologous Groups of proteins) analysis.

### 2.5. Statistical Data Analysis

One-way analysis of covariance (SPSS version 24.0, ANCOVA) in SPSS software was used to analyze the significance of the differences in muscle fatty acids, fatty acid-related gene expression, and the ileum alpha diversity indices (Ace index, Chao1 index, Shannon index, and Simpson index) in Tibetan sheep at different ages, and *p* < 0.05 was taken as *p* < 0.05 was considered statistically significant; Spearman’s correlation test was used to analyze the correlation between muscle fatty acid and fatty acid-related gene expression and the correlation between muscle fatty acid and fatty acid-related gene mRNA expression and genus-level ileum microbiota (relative abundance > 0.5%).

## 3. Results

### 3.1. Analysis of Muscle Fatty Acid Composition of Tibetan Sheep at Different Ages

#### 3.1.1. Muscle SFA Content of Tibetan Sheep of Different Ages

Table 3 shows the compositions and contents of SFAs in the muscle tissues of Tibetan sheep of different ages. Seven SFAs were detected in all three muscle tissues of Tibetan sheep of different ages, with stearic acid (C18:0) being more abundant than all the other SFAs at all muscle tissue sites. In the longest dorsal muscle, the levels of daturic acid (C17:0) and stearic acid (C18:0) were significantly higher in the 1.5-year-old group than in the 6-year-old and 4-month-old groups (*p* < 0.05), and the myristic acid (C14:0) content was significantly higher in the 4-month-old group than in the 1.5-year-old and 6-year-old groups (*p* < 0.05); however, the difference was not significant compared with that in the 3.5-year-old group. In the foreleg muscle, myristic acid (C14:0) and daturic acid (C17:0) levels were significantly higher in the 1.5-year-old group than in the 3.5-year-old group (*p* < 0.05), butyric acid (C4:0) levels were highest in the 3.5-year-old group and significantly higher than in the 1.5-year-old and 6-year-old groups, and tridecanoic acid (C13:0), pentadecanoic acid (C15:0), and docosa-octadecanoic acid (C22:0) levels were highest in the 3.5-year-old group and significantly higher than in the other groups (*p* < 0.05). Group was the highest and was significantly higher than that of the other groups (*p* < 0.05). In the hindleg muscle, butyric acid (C4:0) and tridecanoic acid (C13:0) levels were significantly higher in the 1.5-year-old group than in the other groups (*p* < 0.05), and daturic acid (C17:0) and stearic acid (C18:0) levels were significantly higher in the 6-year-old group than in the 1.5-year-old group (*p* < 0.05).

#### 3.1.2. Muscle UFAs Content in Tibetan Sheep of Different Ages

Determination of the composition and content of UFA in different parts of muscle tissues of Tibetan sheep of different ages revealed that in Table 4, ten UFAs were detected in all three muscle tissues of Tibetan sheep of different ages. The oleic acid (C18:1n9c) content was higher than that of all other UFAs in all muscle tissue sites. In the longest dorsal muscle, the content of cis-10-heptadecenoic acid (C17:1), linoleic acid (C18:2n6c), and arachidonic acid (C20:4n6) were the highest in the 1.5-year-old group, cis-10-heptadecenoic acid (C17:1) content was significantly higher in the 1.5-year-old group than in the other groups (*p* < 0.05), cis-5,8,11,14,17-eicosapentaen-oic acid (C20:5n3) content was significantly higher in the 1.5-year-old group than in the 3.5-year-old and the 4-month-old groups (*p* < 0.05), and elaidic acid (C18:1n9t) content was significantly higher in the 3.5-year-old group than in the other groups (*p* < 0.05). In the foreleg muscle, the content of cis-10-heptadecenoic acid (C17:1), oleic acid (C18:1n9c), and docosahexaenoic acid (C22:6n3) were the highest in the 1.5-year-old group, while the contents of myristoleic acid (C14:1) and linoleic acid (C18:2n6c) were significantly higher in the 3.5-year-old group (*p* < 0.05); the contents of arachidonic acid (C20:4n6) and cis-5,8,11,14,17-eicosapentaen-oic acid (C20:5n3) were highest in the 4-month-old group, significantly higher (*p* < 0.05) than that in the 1.5- and 6-year-old groups, but the difference from that in the 3.5-year-old group was not significant; and the contents of elaidic acid (C18:1n9t) and alpha linoleic acid (C18:3n3) were significantly (*p* < 0.05) higher (*p* < 0.05) in the 6-year-old group than in the other groups. In the hindleg muscles, the contents of myristoleic acid (C14:1), methyl linoleate (C18:2n6c), and alpha linoleic acid (C18:3n3) and docosahexaenoic acid (C22:6n3) were significantly higher in the 1.5-year-old group than in the other groups (*p* < 0.05), and the contents of arachidonic acid (C20:4n6) and cis-5,8,11,14,17-eicosapentaen-oic acid (C20:5n3) were significantly higher (*p* < 0.05) than in the 3.5- and 6-year-old groups, but the difference between the content in these groups and that in the 4-month-old group was not significant (*p* > 0.05). The content of elaidic acid (C18:1n9t) was significantly higher (*p* < 0.05) in the 6-year-old group than in the 1.5-year-old group but not significantly different (*p* > 0.05) from that in the 3.5-year-old group.

#### 3.1.3. Total Fatty Acid Content of Muscle in Tibetan Sheep of Different Ages

Determination of the composition and content of total fatty acids in different parts of muscle tissues of Tibetan sheep of different ages revealed that, as shown in Table 5, the UFA content was higher than the SFA content, and the MUFA content was higher than the PUFA content. In the longest back muscle, the PUFA content was highest in the 4-month-old and 1.5-year-old groups; the SFA content was highest in the 3.5-year-old age group and was significantly higher than that in the other age groups (*p* < 0.05). In the front leg muscles, the MUFA content was significantly higher (*p* < 0.05) in the 1.5-year-old group than in the 3.5-year-old group. In the hindleg muscle, the PUFA content was significantly higher in the 1.5-year-old group than in the other groups (*p* < 0.05), and the SFA content was significantly higher in the 3.5-year-old group than in the 1.5-year-old and 4-month-old groups (*p* < 0.05); the PUFA-to-SFA ratio in the different muscle tissues (P/S) ranged from 0.280–0.798, and the P/S in the hindleg muscle was the highest in the 1.5-year-old group and was significantly higher than that in the other groups (*p* < 0.05). In the foreleg muscle, the P/S ratio was significantly higher in the 3.5-year-old group than in the other groups (*p* < 0.05).

### 3.2. Expression Characteristics of Muscle Quality Related Genes of Different Ages in Tibetan Sheep

By performing quantitative real-time fluorescence detection of genes related to longest dorsal muscle and leg muscle quality in Tibetan sheep of different ages, it was found that in the longest dorsal muscle Figure 1a, *LPL* gene expression was significantly higher in the 1.5-year-old group than in the 4-month-old and 3.5-year-old groups (*p* < 0.05); *SCD* and FABP4 gene expression was significantly higher in the 1.5-year-old group than in the 4-month-old group but was significantly lower than in the 3.5-year-old group (*p* < 0.05), with *FABP4* gene expression being significantly higher in the 1.5-year-old group than in the 6-year-old group (*p* < 0.05); and *FASN* and *PPAR* gene expression was significantly higher in the 6-year-old group than in the other groups (*p* < 0.05). As shown in Figure 1b, the *LPL* and *FABP4* gene expression levels were significantly higher in the 6-year-old group (*p* < 0.05), the *SCD* and *PPAR* gene expression levels were significantly higher in the 1.5-year-old group than in the 4-month-old and 6-year-old groups (*p* < 0.05), and *FASN* gene expression was significantly higher in the 4-month-old group than in the 1.5-and 6-year-old groups (*p* < 0.05).

### 3.3. Characteristics of Ileum Microbiota of Different Ages in Tibetan Sheep

#### 3.3.1. The Ileum Microbiota Diversity

A total of 1,664,927 pairs of reads were obtained from the study, and a total of 1,419,768 clean reads were generated after double-ended reads quality control and splicing, with each sample generating at least 54,603 clean reads and an average of 70,988 clean reads. The reads were clustered using Usearch software (version 10.0) at the 97% similarity level, and a total of 21,187 OTUs were obtained; of these, 7828, 425, 5688, and 7483 OTUs were found in the 4-month-old, 1.5-year-old, 3.5-year-old, and 6-year-old age groups, respectively. There were 270 OTUs common to the ileum microbiota from different age groups, with the highest number of OTUs being unique to the 1.5-year-old group. The highest number of OTUs was 6111 (Figure 2a). Dilution curves depicted species diversity and species richness across samples, with the curves leveling off at 20,000 reads, indicating saturation of sequencing coverage (Figure 2b). The Shannon index reflects the microbiota diversity of each sample at different sequencing numbers, and the curve flattens out at 20,000 reads, indicating that the sequencing number of sequences was large enough to cover the vast majority of microbiota species. As shown in Figure 3, alpha index diversity analysis revealed that the coverage of this sequence reached more than 99.93%, and the sequencing data fully reflected the authenticity and reliability of the samples. Specifically, the Shannon, Chao1, and ACE indices were the highest in the 1.5-year-old age group and were significantly higher than those in the 3.5-year-old age group (*p* < 0.05); moreover, the difference from the other age groups was not significant.

PCoA analysis revealed that the clustering of samples in different subgroups was obvious and that there were significant differences between the different subgroups. The ileum microbiota diversity of the 1.5-year-old group differed significantly from that of the other groups, and the ileum microbiota diversity of the 6-year-old age group was similar to that of the 3.5-year-old age group but differed significantly from that of the 4-month-old group (Figure 2c). Anosim analysis revealed that the between-group differences were significantly greater than the within-group differences (*p* < 0.05), and the sequencing data were highly reliable for subsequent analyses (Figure 2d).

#### 3.3.2. The Ileum Microbiota Species Composition

At the taxonomic level, 43 phyla, 104 classes, 277 orders, 584 families, 1170 genera, and 1411 species were detected. At the phylum level, Bacteroidetes, Firmicutes, Patescibacteria, Proteobacteria, and Actinobacteria were the dominant phyla, with relative abundances greater than 1% (Figure 4a). The proportion of thick-walled phyla was the highest of all four age groups, accounting for more than 50% cent of the total abundance, and this proportion was significantly (*p* < 0.05) higher in the 1.5-year-old group than in the 4-month-old group. The Anaplasma phylum had the highest abundance in the 1.5-year-old group, which was significantly higher than that in the 3.5-year-old and 6-year-old groups (*p* < 0.05). The proportion of Anaplasma was significantly higher (*p* < 0.05) in the 1.5-year-old group than in the 4-month-old and 6-year-old groups but significantly lower (*p* < 0.05) than in the 3.5-year-old group. Further analysis based on rank sum test plots revealed four intergroup differential species at the phylum level; the thick-walled phylum with the highest abundance did not significantly differ across time; among these species, the abundance of Cyanobacteria was significantly lower in the 1.5-year-old group than in the 3.5-year-old group (*p* < 0.05), the abundance of Desulfobacterota was significantly higher in the 4-month-old group than in the 3.5-year-old group (*p* < 0.05), the abundance of Proteobacteria was significantly higher in the 3.5-year-old group than in the 6-year-old group (*p* < 0.05), and the difference in abundance of these bacteria between the 3.5-year-old group and the 1.5-year-old group was not significant.

At the genus level *Romboutsia*, *unclassified_Lachnospiraceae*, *uncultured_rumen_bacterium, Aeriscardovia*, *unclassified_[Eubacterium]_coprostanoligenes group*, and *NK4A214_group* were the dominant genera, and the abundance ratio of *NK4A214_group* was significantly higher in the 3.5-year-old and 6-year-old groups than in the remaining two age groups (*p* < 0.05); *Escherichia_Shigella Shigella*, *Clostridium_sensu_stricto_1 Clostridium* spp., *uncultured_rumen_bacterium* and *unclassified_Peptostreptococcaceae* had significantly higher abundance ratios in the 3.5-year-old age group than in the 1.5-year-old age group (*p* < 0.05); *Romboutsia*, *unclassified_ Lachnospiraceae*, *unclassified_[Eubacterium]_coprostanoligenes_group*, and *Christensenellaceae_R_7_group* had significantly higher abundance proportions in the 1.5-year-old age group than in the remaining age groups (*p* < 0.05) (Figure 4b).

#### 3.3.3. Analysis of Differential Microbiota of the Ileum at Different Ages

In order to find the Biomarker with statistically significant differences between different groups, LEFSE (Line Discriminant Analysis (LDA) Effect Size) analysis was performed at the genus level, and as can be seen in Figure 5, there were more differentiated groups of bacteria in the 1.5-year-old group, with *p_Bacteroidota*, *c_Bacteroidia*, *0_Bacteroidales* were the more differentiated genera in the 1.5-year-old group; *c_Gammaproteobacteria*, *g_Escherichia_Shigella*, *s_unclassified_Escherichia_Shigella*, *f_Enterobacteriaceae*, *o_Enterobacterles*, *p_Proteobacteria* for the more differentiated genera in the 3.5-year-old group, *p_Actinobacteriota* for the more differentiated genera in the 4-month-old group, *o_Peptostreptococcales_Tissierellales*, *f_Peptostreptococcaceae* were the more differentiated genera in the 6-year-old group.

#### 3.3.4. Prediction of the Ileum Microbiota Function

Gene function prediction of the ileum microbiota 16S rRNA sequencing data using PICRUSt software revealed a total of 46 KEGG gene families and 25 COG functional genes in the 1.5-year-old and 4-month-old age groups in the 1.5-year-old and 3.5-year-old age groups, in the 3.5-year-old and 6-year-old age groups, and in the 4-month-old and 6-year-old age groups, with more than 26% and more than 1% of genes being associated with the METABOLISM pathway, respectively. Genes were more than 26.1% and 32%, respectively. A total of five differentially functional genes were identified by differential analysis of KEGG metabolic pathways between the 1.5-year-old and 4-month-old groups (Figure 6a), with no significant differences in any of the METABOLISM pathway genes, and no differentially functional genes were identified by differential analysis of COG function prediction. A total of 11 differentially functional genes were identified by KEGG metabolic pathway difference analysis between the 1.5-year-old and 3.5-year-old groups (Figure 6b), with 55% belonging to the METABOLISM pathway, where biosynthesis of other secondary metabolites, glycan biosynthesis and metabolism and global and overview maps were significantly higher in the 1.5-year-old group than in the 3.5-year-old group (*p* < 0.05). Differential analysis of COG function prediction (Figure 7b) identified nine differentially functional genes, with coenzyme transport and metabolism and lipid transport and metabolism in the METABOLISM pathway being significantly higher in the 1.5-year-old group than in the 3.5-year-old group (*p* < 0.05). A total of 23 differentially functional genes were identified by KEGG metabolic pathway difference analysis between the 3.5-year-old and 6-year-old groups (Figure 6c), of which carbohydrate metabolism, lipid metabolism, xenobiotics biodegradation, and metabolism and metabolism of other amino acids were significantly higher (*p* < 0.05) in the 3.5-year-old group than in the 6-year-old group, no differentially functional genes were identified by differential analysis of COG function prediction. A total of 19 differentially functional genes were identified by differential analysis of KEGG metabolic pathways between the 4-month-old and 6-year-old groups (Figure 6d), with 26% belonging to the METABOLISM pathway, with global and overview maps being significantly higher at 4-month-old age than in the 6-year-old group (*p* < 0.05). A total of nine differentially functional genes were identified by differential analysis of COG functional prediction Figure 7a), with 33% belonging to the METABOLISM pathway, with the highest proportion of amino acid transport and metabolism being significantly higher in the 4-month-old group than in the 6-years-old group (*p* = 0.012).

### 3.4. Analysis of the Correlation between Muscle Fatty Acids and Related Gene Expression in Tibetan Sheep of Different Ages

By analyzing the correlation between muscle fatty acids and their related genes in Tibetan sheep of different ages, it was found that there was a correlation between muscle fatty acids and their related genes in Tibetan sheep. Among them, in the longest dorsal muscle (Figure 8a), butyric acid (C4:0), oleic acid (C18:1n9c), alpha linoleic acid (C18:3n3), arachidonic acid (C20:4n6), cis-5,8,11,14,17-eicosapentaen-oic acid (C20:5n3), and docosahexaenoic acid (C22:6n3) were significantly and positively correlated with the expression of the *LPL* gene (*p* < 0.05); myristic acid (C14:0) and stearic acid (C18:0) were significantly positively correlated with *SCD* gene expression (*p* < 0.05); myristic acid (C14:0) was negatively correlated with *FASN* gene expression and the difference was not significant (*p* > 0.05); pentadecanoic acid (C15:0), daturic acid (C17:0), stearic acid (C18:0) and docosahexaenoic acid (C22:6n3) were significantly positively correlated (*p* < 0.05); oleic acid (C18:In9c) and arachidonic acid (C20:4n6) were significantly positively correlated (*p* < 0.05) with *FABP4* gene expression.

In the hamstrings (Figure 8b), oleic acid (C18:1n9c) and arachidonic acid (C20:4n6) were significantly and positively correlated (*p* < 0.05) with the *LPL*, *SCD*, and *FABP4* gene expression. Myristic acid (C14:0) and stearic acid (C18:0) showed a significant positive correlation (*p* < 0.05) with the *LPL* and *PPAR* gene expression. Cis-5,8,11,14,17-eicosapentaen-oic acid (C20:5n3) showed a significant positive correlation (*p* < 0.05) with *LPL* and *FASN* gene expression. *LPL* gene expression was negatively correlated with pentadecanoic acid (C15:0) and docosahexaenoic acid (C22:6n3) with a non-significant difference (*p* > 0.05). Pentadecanoic acid (C15:0), daturic acid (C17:0), *SCD*, and *PPAR* gene expression were significantly positively correlated (*p* < 0.05).

### 3.5. Analysis of the Ileum Microbiota-Muscle Fatty Acid-Realted Gene Associations in the Tibetan Sheep

As shown in Figure 9, correlation heatmaps were constructed between the ileum microbiota (genus level microbes in the top 20 relative abundance) and muscle fatty acids and muscle fatty acid-related genes in Tibetan sheep (correlation threshold > 0.5). Muscle fatty acids were found to be significantly correlated (*p* < 0.05) with 39 genus level microbiota, 23 positively and 16 negatively. *Brevinema*, *unclassified_F082*, *unclassified_UCG_010*, *Christensenellaceae_R_7_group*, *UCG_005* were significantly and positively correlated with arachidonic acid (C20:4n6), docosahexaenoic acid (C22:6n3), and cis-5,8,11,14,17-eicosapentaen-oic acid (C20:5n3); *unclassified_F082*, *unclassified_UCG_010*, *Christensenellaceae_R_7_group*, and *UCG_005* were significantly and positively correlated with arachidonic acid (C20:4n6) (*p* < 0.05), with *unclassified_F082* being highly significantly positively correlated (*p* < 0.01) with arachidonic acid (C20:4n6). *Romboutsia*, *unclassified_Peptostreptococcaceae* were significantly positively correlated (*p* < 0.05) with oleic acid (C18:1n9c), with the *Romboutsia* being highly significantly positively correlated (*p* < 0.01) with it. The *Romboutsia* was significantly positively correlated (*p* < 0.05) with alpha linoleic acid (C18:3n3), *Romboutsia* was significant positive correlation (*p* < 0.05), *Clostridium_sensu_stricto_1*, *Escherichia_Shigella* were significantly positively correlated (*p* < 0.05) with stearic acid (C18:0). *Candidatus_Saccharimonas* was significantly positively correlated (*p* < 0.05) with alpha linoleic acid (C18:3n3), myristic acid (C14:0) were significantly positively correlated (*p* < 0.05), and *uncultured_rumen_bacterium* was highly significantly positively correlated (*p* < 0.01) with myristic acid (C14:0). *unclassified_[Eubacterium]_coprostanoligenes_group* showed a highly significant positive correlation (*p* < 0.05) with linoleic acid (C18:2n6c). *Brevinema*, *unclassified_UCG_010*, *Christensenellaceae_R_7_group*, and *UCG_005* showed a highly significant negative correlation (*p* < 0.01) with stearic acid (C18:0). *Uncultured_rumen_bacterium* showed highly significant negative correlation (*p* < 0.01) with cis-5,8,11,14,17-eicosapen-taen-oic acid (C20:5n3). *Turicibacter*, *Clostridium_sensu_stricto_1* showed a highly significant negative correlation (*p* < 0.01) with arachidonic acid (C20:4n6). *Turicibacter*, *Clostridium_sensu_stricto_1*, and *uncultured_rumen_bacterium* were significantly negatively correlated (*p* < 0.05) with docosahexaenoic acid (C22:6n3).

In addition, genes associated with muscle fatty acids were correlated with 12 genus-level microbiota. Among them, *unclassified_[Eubacterium]_coprostanoligenes_group* showed a highly significant positive correlation with *SCD* and *FABP4* genes (*p* < 0.01), and *Escherichia_Shigella* showed a highly significant positive correlation with *FASN* gene (*p* < 0.01). The *Romboutsia*, *unclassified_Peptostreptococcaceae* showed a highly significant negative correlation (*p* < 0.01) with the *FABP4* and *SCD* genes, *unclassified_F082* showed a significant negative correlation (*p* < 0.05) with *FASN* gene, *Brevinema*, *Romboutsia, unclassified_Peptostreptococcaceae* and *LPL* genes were significantly negatively correlated (*p* < 0.05).

## 4. Discussion

With the continuous improvement in people’s living standards, people’s consumption demand for meat has gradually shifted from a quantitative demand to a qualitative demand. The flavor characteristics of meat have a direct impact on consumers’ desire to buy meat, and livestock products are an important source of fatty acids for human beings. Research has shown that the content and composition of muscle fatty acids are important indicators of meat quality [30]. The richness and content of fatty acids and their content are the main characteristics of Tibetan sheep as a high-quality meat. The nutritional value of meat is generally measured in terms of PUFA/SFA (P/S), which is considered to be favorable to human health when it is at a value of 0.4 or slightly higher (but not too high) [31,32]. In the present study, the P/S values of the longest dorsal muscle were higher than 0.414 in both 4-month-old and 1.5-year-old Tibetan sheep, and the P/S values of the hindleg muscles were higher than 0.4 in all four age groups and were significantly higher in the 1.5-year-old group. These results are in accordance with the nutritional values of Wood and Banskalieva [33] and others and are more in line with today’s human nutritional regulations. Lamb also contains 45–50% of SFAS, such as myristic acid (C14:0) and stearic acid (C18:0). It has been suggested that increased SFA intake promotes adipose tissue expansion and hypertrophy, leading to apoptosis, which may increase the risk of cardiovascular disease and metabolic syndrome [34]. In the present study, the SFA content in the longest back muscle was significantly higher in the 3.5-year-old group than in the other age groups, with myristic acid (C14:0) content being the highest in the 3.5-year-old group. Similarly, in the anterior and posterior hamstrings, the myristic acid (C14:0) content was higher in the 6-year-old group than in the other age groups, suggesting that the meat quality of the 1.5-year-old group was more favorable to human health than the 3.5-year-old and 6-year-old groups. Second, the content of stearic acid (C18:0) in the longest dorsal muscle was significantly higher in the 3.5-year-old group, and that in the anterior and posterior leg muscles was higher in the 6-year-old group, suggesting that sheep meat becomes less tasty with age [35], leading to a decrease in meat quality [36]. Compared to SFAs, MUFAs provide beneficial effects, such as anti-cancer, anti-autoimmune disease, and anti-inflammatory disease effects, by lowering total and LDL cholesterol in the blood without destroying beneficial HDL cholesterol [37,38]. Cameron and Enser [39], reported that MUFA content is positively correlated with the flavor of meat and thus influences the acceptability of meat. Oleic acid (C18:1n9c), the most important MUFA in sheep fat, has a positive effect on the absorption of other fatty acids [40]. In this study, the MUFA content in the longest back muscle was higher at all four ages, and in the foreleg muscle, the MUFA content was highest in the 1.5-year-old group and significantly higher than in the 3.5-year-old group. These findings indicate that the longest back muscle contained the greatest amount of beneficial components and the best quality meat among the three muscle tissues and that the 1.5-year-old group had more beneficial components for the human body than the 3.5-year-old group. Oleic acid (C18:1n9c) was the most abundant compound in all three muscle tissues and was more abundant in the longest back muscle in the 1.5-year-old group than in the 3.5-year-old group. Reducing the intake of SFAs while increasing the intake of PUFAs reduces the incidence of cardiovascular disease in humans [6,41]. In this study, the PUFA content in the hindleg muscle was significantly higher in the 1.5-year-old group than in the other groups. The PUFAs arachidonic acid (C20:4n6) and alpha linoleic acid (C18:3n3) were found to be more abundant in the 1.5-year-old group in both the longest dorsal muscle and hindleg muscle, and these PUFAs are effective at lowering the accumulation of cholesterol and inhibiting tumor incidence, which is beneficial to consumer health [42]. In addition, the content of the omega-3 PUFAs cis-5,8,11,14,17-eicosapentaen-oic acid (C20:5n3) and docosahexaenoic acid (C22:6n3) (DHA) were higher in both the longest dorsal muscle and hind leg muscle in the 1.5-year-old group than in the other groups, and cis-5,8,11,14,17-eicosapentaen-oic acid (C20:5n3) can play a positive role in the treatment of cardiovascular diseases such as atherosclerosis [43] and docosahexaenoic acid (C22:6n3); this PUFA is often referred to as brain gold, as it can reduce platelet aggregation and lipid abundance, prevent coronary heart disease, improve memory and prevent brain aging [44,45]. Therefore, from an overall nutritional perspective, the older the lamb becomes, the lower its fatty acid value. Moreover, changes in fatty acids during heat treatment should be noted. Heat treatment can increase the content of unsaturated fatty acids and improve the utilization value of oil. Several studies have shown that heat treatment is considered an effective denaturant that promotes the decomposition of proteins and can reduce their potential sensitization [46]. However, high-temperature cooking will increase the content of trans fatty acids, and the intake of too many trans fatty acids will cause a series of adverse effects on the human body. Therefore, it has been proposed that during heat treatment, oven baking is the best heat treatment for preserving all the lipid characteristics of meat, including the PUFA content and n-3/n-6 ratio [47].

Although differences in meat quality are closely related to gene expression, the extent to which the molecular mechanisms leading to differences in meat quality traits among Tibetan sheep of different ages are based on gene expression is unclear. In the present study, the longest dorsal muscle of the 1.5-year-old group showed high expression of the *FABP4* gene, which is responsible for transporting fatty acids from the cell membrane to fatty acid utilization sites in the cell [48]. Studies have reported that the expression of the *LPL* and *FABP4* genes is related to fat content and meat quality to some extent [49,50] and that the *LPL* gene is an important factor in assessing meat quality. In this study, *LPL* gene expression in the longest back muscle was high in the 1.5-year-old group resulting in an increase in fat content, and the *LPL* gene in the hamstring muscle regulated the content of stearic acid (C18:0), resulting in an increase in its content in the 6-year-old group, and an increase in the foul flavor of mutton. In addition, the *FABP4* gene regulates the increase in oleic acid (C18:1n9c) and arachidonic acid (C20:4n6) content in the hamstrings. The *FASN* gene has an important role in fat deposition, whereas the *SCD* and *PPAR* genes play a role in lipid metabolism [51,52]. The expression of the *FASN* and *PPAR* genes increased with age, whereas the *PPAR* gene regulated the increased expression of myristic acid (C14:0) and stearic acid (C18:0), i.e., the older the grazing Tibetan sheep were, the higher the expression of the *PPAR* gene, which led to the increase in the content of some of the harmful fatty acids. The *SCD* gene regulated the high expression of cis-5,8,11,14,17-eicosapentaen-oic acid (C20:5n3) and arachidonic acid (C20:4n6), which were highly expressed. The *FASN*, *SCD*, and *PPAR* genes were highly expressed in the hamstrings of the 3.5-year-old group, which was similar to the findings of Wang et al.’s study on Laiwu pigs [53]. However, the differences in the expression of these genes in different parts of the body need to be further investigated.

In addition, it was found that the gastrointestinal microbiota is involved in biohydrogenation and isomerisation in vivo and influences muscle fatty acid deposition through the fatty acid composition thereof [17]. The diversity of the gut microbiota in animals changes significantly with age, and this process is influenced by a number of factors, including feed and genetics [54,55]. In the present study, the ileum microbiota diversity was significantly greater in the 1.5-year-old group than in the control group, and the proportions of *mycobacterium anisopliae* and *mycobacterium thickeniophilum* were also higher in the 1.5-year-old group. These findings are more favorable for nutrient uptake and utilization by hosts [56], which is similar to the findings of previous studies [57,58]. At the genus level, *Rikenellaceae* (*RC9_gut_group*) was the dominant bacterium in the 1.5-year-old group, and it has been suggested that this group is closely related to members of the Alistipes family and may play a role in degrading plant-derived polysaccharides [59,60]. It was hypothesized that the degradation of cellulosic polysaccharides in Tibetan sheep was higher in the 1.5-year-old group than in the control group and could provide more energy to the body. Many cellulolytic bacteria such as *Ruminalococcus* spp., *Fibrobacter* spp., *Vibrio butyric acidicus* spp., *Spirochaetes densiflorus* spp., *Vibrio pseudobutyric acidicus* spp. were also identified in this study at the four ages, and the *fiber-degrading bacteria* are an important group of bacteria that play a key role in the degradation of cellulosic substances to produce VFAs [61,62]. Among them, *Vibrio* spp. *Butyrate*, *Vibrio densely spirochete* spp. and Vibrio pseudobutyrate exhibited large differences in relative abundance at different ages, which may be related to methane production, and a positive correlation between the number of *Vibrio* spp. and the number of methanogens has been reported in some studies [63], whereas *Vibrio* spp. butyrate was significantly higher in the 3.5-year-old group in the present study. KEGG co-annotation analysis revealed that the differentially expressed metabolites were mainly annotated as being involved in amino acid metabolism, lipid metabolism, and the digestive system. Amino acids in the gastrointestinal tract are key precursors for protein and peptide synthesis. These amino acids are derived mainly from dietary proteins and microproteins [64] and are subsequently absorbed and transported to participate in muscle protein synthesis. In addition, lipid metabolism in the gastrointestinal tract is highly active [65] and is involved mainly in regulating the antimicrobial effects of fatty acids and hydrogenation by the microbiota, altering fatty acid uptake and thus improving productive performance [66]. In the present study, the microbiota in the ileum may influence fatty acid metabolism in muscle through lipid metabolism, thus affecting fat deposition (Figure 10).

Therefore, in the present study, the microbiota was correlated with muscle fatty acids and host gene expression, and *Romboutsia*, and *unclassified_Peptostreptococcaceae* were found to be significantly correlated with oleic acid (C18:1n9c). *Romboutsia* species have been demonstrated to produce VFAs, which can ameliorate metabolic endotoxemia and protect the intestinal barrier [67,68], and play an important role in intestinal health [69]. Moreover, *Romboutsia* was negatively correlated with *SCD* gene expression, resulting in a high level of stearic acid in the 3.5-year-old group and a lower level in the 1.5-year-old group. In addition, *Clostridium_sensu_stricto_1* showed a highly significant negative correlation with arachidonic acid (C20:4n6), a significant positive correlation with stearic acid (C18:0), and a positive correlation with the expression of the *FASN* gene in the 3.5-year-old group. *Clostridium_sensu_stricto_1* has been reported to be a conditional pathogen that induces intestinal diseases. In addition, *Christensenellaceae_R_7_group* was also significantly positively correlated with arachidonic acid (C20:4n6) and negatively correlated with stearic acid (C18:0), and the proportion of *Christensenellaceae_R_7_group* was significantly higher in the 1.5-year-old group than in the other groups; moreover, the proportion of *Christensenellaceae_R_7_group* belongs to the phylum thick-walled bacteria [70], which mainly breaks down fibrous materials [71], and suggests that the Tibetan sheep in the 1.5-year-old group breaks down more fibrous materials in the ileum microbiota, which provides more energy for the organism. *Escherichia Shigella* has been shown to aggravate alcoholic cirrhosis [72], whereas in Tibetan sheep *Escherichia_Shigella* was significantly correlated with stearic acid and *FASN* gene expression in the 3.5-year-old group. In the present study, *unclassified_[Eubacterium]_coprostanoligenes_group* was significantly associated with linoleic acid (C18:2n6c), and *unclassified_ [Eubacterium]_coprostanoligenes_group* was highly significantly positively associated with *SCD*, while the *FABP4* gene was highly significantly positively correlated with these factors. *FABP4* was found to be significantly correlated with palmitic acid [73], confirming the involvement of the *FABP4* gene in fatty acid metabolic processes. In conclusion, there is a relationship between microbiota metabolites and muscle fatty acid metabolites.

## 5. Conclusions

With increasing age, more adverse fatty acids such as myristic acid (C14:0) and stearic acid (C18:0) were deposited. In the 1.5-year-old group, the high expression of the *LPL*, *SCD*, and *FABP4* genes increased the content of PUFAs in Tibetan sheep meat, decreased the content of unfavorable fatty acids, and the content of PUFAs in the longissimus dorsi muscle was higher than that in the leg muscle, indicating that 1.5-year-old Tibetan sheep meat was more suitable for consumption and that the longissimus dorsi muscle was more suitable than the leg muscle. The differentially abundant microbial metabolites were related mainly to lipid metabolism and the digestive system, and the microbial community was significantly related to meat quality, muscle fatty acids and gene expression. Due to the hydrogenation of ileal microorganisms, the microbiota regulates the content and composition of muscle fatty acids, thus affecting muscle fat deposition. Therefore, it is recommended that the optimal slaughter age for Tibetan sheep be 1.5 years.

## Figures and Tables

**Figure 1 foods-13-00679-f001:**
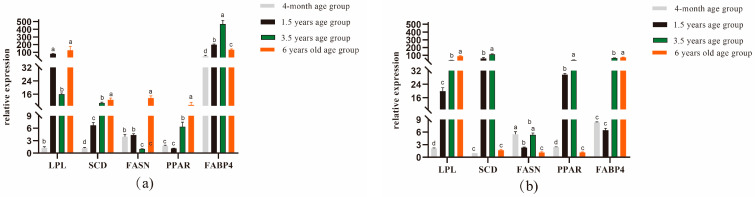
Expression analysis of genes related to dorsal longest muscle and leg muscle in Tibetan sheep of different ages. (**a**) Dorsal longest muscle gene expression analysis (**b**) Leg muscle gene expression analysis. Note: different lowercase letters indicate significant differences between ages at the *p* < 0.05 level.

**Figure 2 foods-13-00679-f002:**
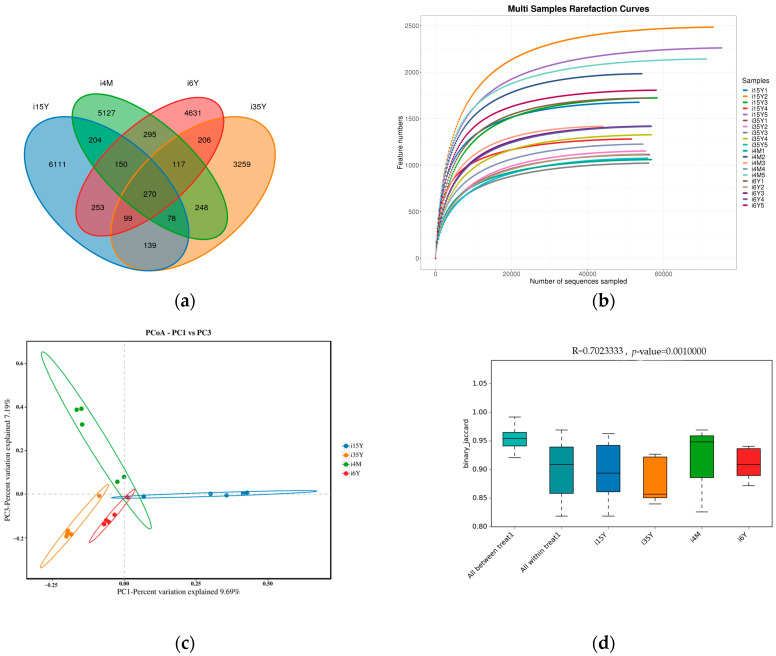
Analysis of ileum microbiota diversity. (**a**) OTU Venn diagram analysis by age; (**b**) dilution curve analysis; (**c**) PCoA analysis; (**d**) Anosim analysis.

**Figure 3 foods-13-00679-f003:**
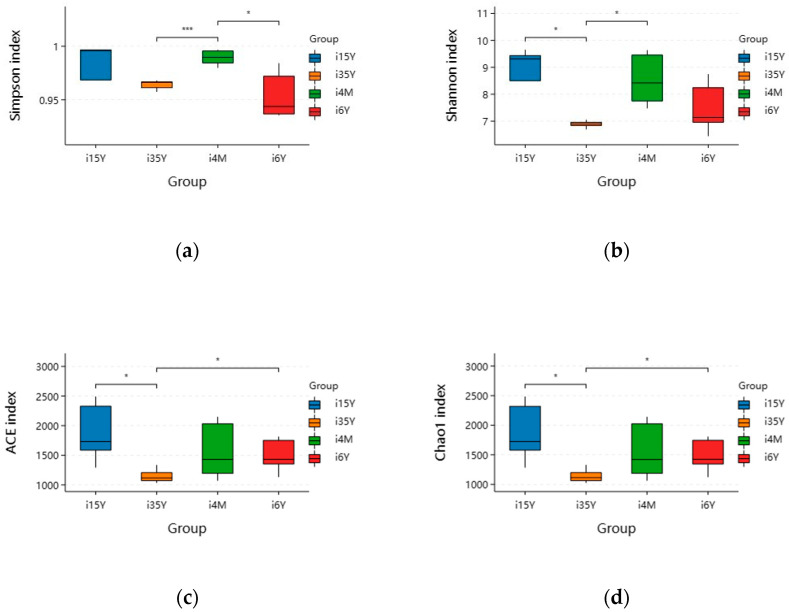
Analysis of ileum microbiota diversity of different ages in Tibetan sheep. (**a**) Simpson; (**b**) Shannon; (**c**) ACE; (**d**) Chao1. Note: i4M represents the 4-month age group; i15Y represents the 1.5-year age group; i35Y represents the 3.5-year age group; and i6Y represents the 6-year age group. (* *p* < 0.05, *** *p* < 0.001). The same is below.

**Figure 4 foods-13-00679-f004:**
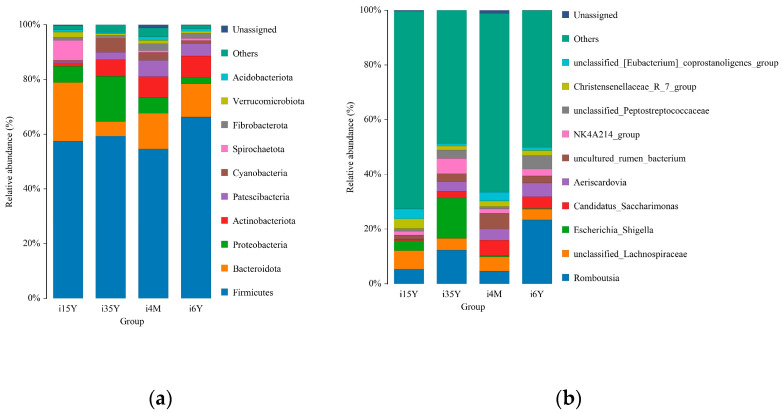
Analysis of species composition. (**a**) Species composition at the phylum level; (**b**) Species composition at the genus level.

**Figure 5 foods-13-00679-f005:**
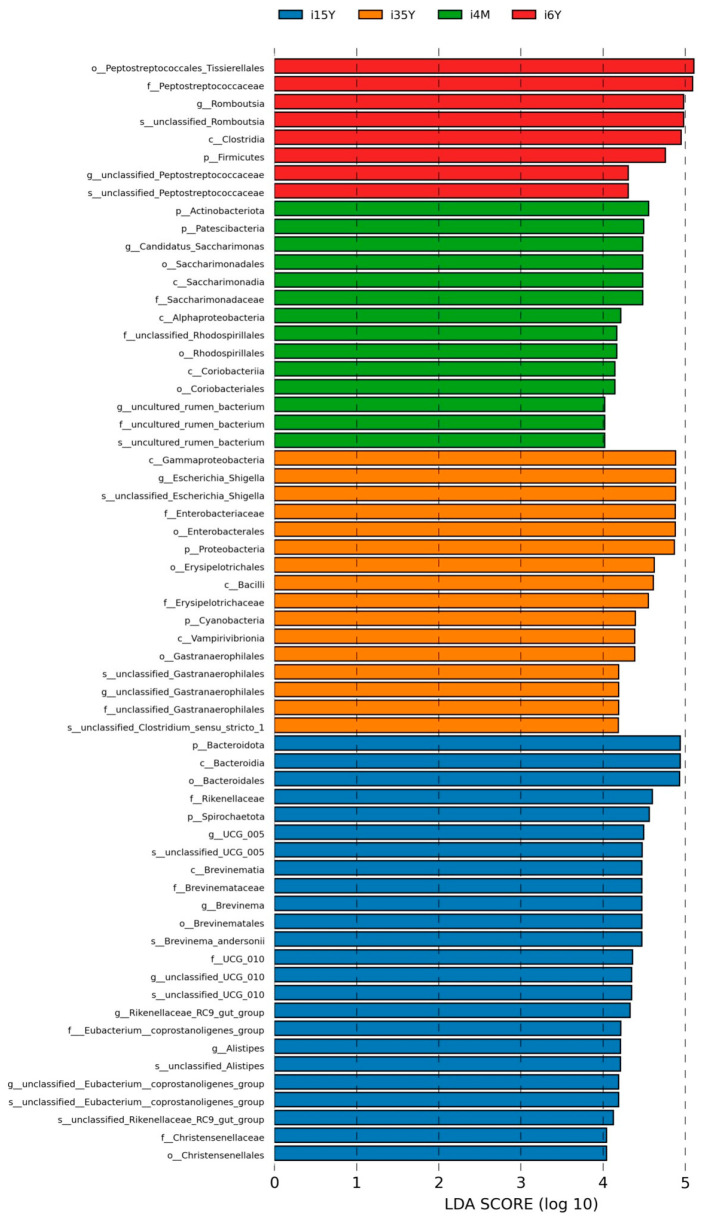
LEFSe analysis.

**Figure 6 foods-13-00679-f006:**
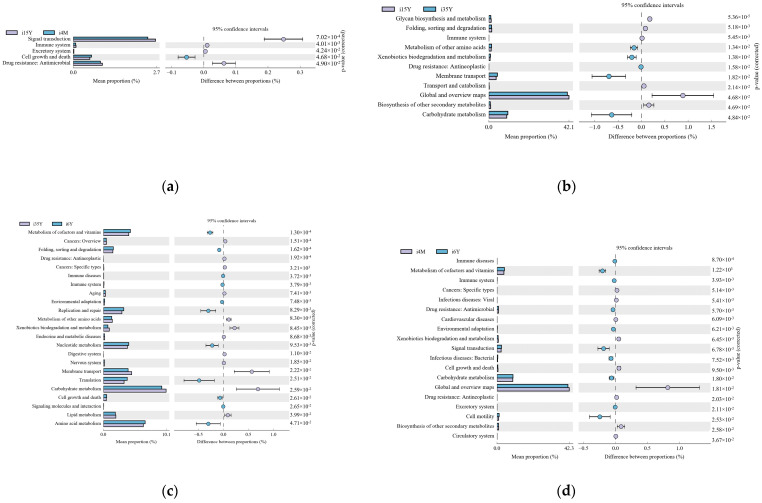
Figure functional prediction of the KEGG gene family. (**a**) 1.5-year-old age group and 4-month-old age group; (**b**) 1.5-year-old age group and 3.5-year-old age group; (**c**) 3.5-year-old age group and 6-year-old age group; (**d**) 4-month-old age group and 6-year-old age group.

**Figure 7 foods-13-00679-f007:**
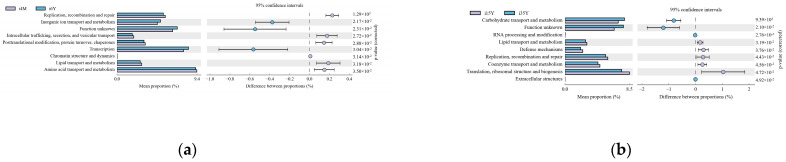
Functional prediction of the COG gene family. (**a**) 4-month-old age group and 6-year-old age group; (**b**) 1.5-year-old age group and 3.5-year-old age group.

**Figure 8 foods-13-00679-f008:**
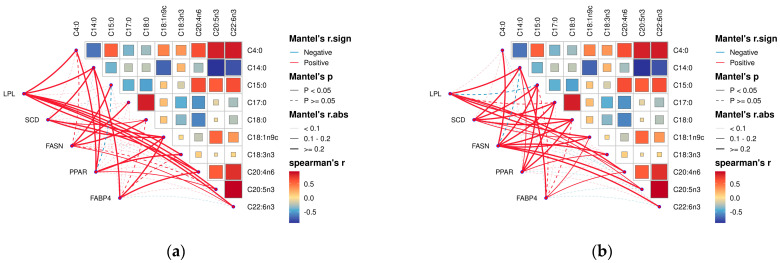
Correlation analysis of dorsal longest muscle and leg muscle of Tibetan sheep of different ages and their muscle fatty acids with related genes. (**a**) Heat map of the relationship between fatty acid content and composition in longissimus dorsi muscle and related genes in leg muscle. (**b**) Heat map of the relationship between fatty acid content and composition in leg muscle and related genes in leg muscle. Note: blue segments indicate negative correlation, red lines indicate positive correlation, solid lines indicate *p* < 0.05, dashed lines indicate *p* ≥ 0.05.

**Figure 9 foods-13-00679-f009:**
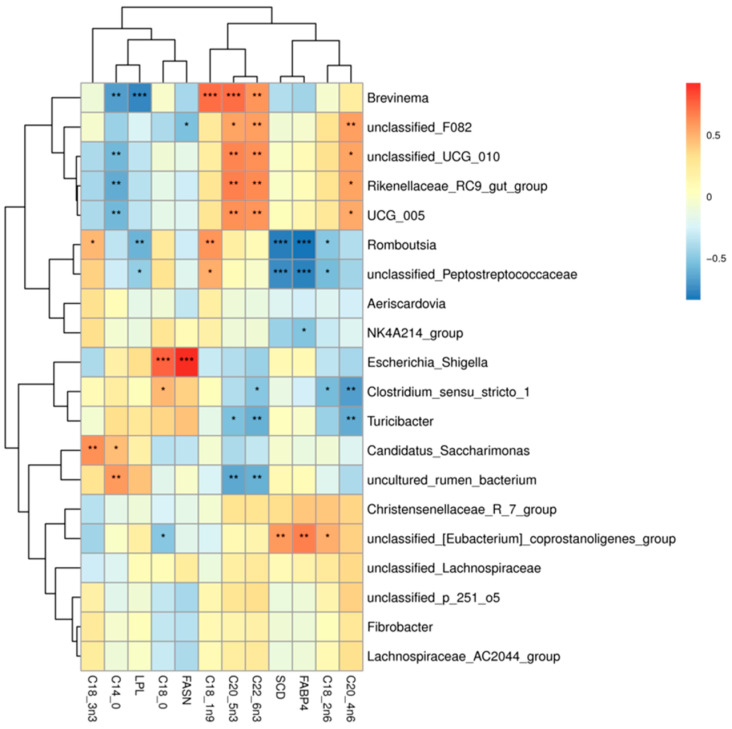
Microbiome-muscle fatty acids-muscle related gene correlation heat map. Note: * *p* < 0.05, ** *p* < 0.01, *** *p* < 0.001.

**Figure 10 foods-13-00679-f010:**
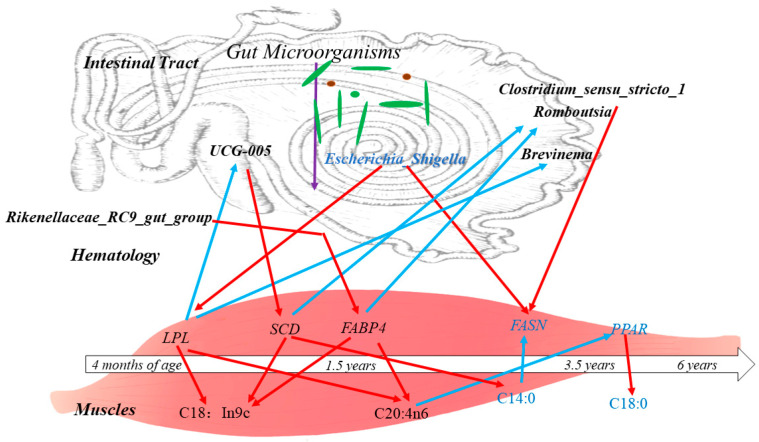
Correlations between the ileum microbiota affecting muscle-related genes and regulating muscle fatty acid composition and content in Tibetan sheep of different ages. Note: black letters indicate upward adjustments, blue letters indicate downward adjustments, red arrows indicate positive correlations and blue arrows indicate negative correlations.

**Table 1 foods-13-00679-t001:** Forage species and nutrient levels [28].

Nutrients (DM Basis)		Dominant Forage Species
CP (%)	10.06	*Poa pratensis* L.
CP (%)	3.77	
Ash (%)	4.55	*Elymus nutans* Griseb
NDF (%)	70.11	
NDF (%)	36.17	*Agropyron cristatum* (L.) Gaertn
HCEL (%)	33.94	
Ca (%)	11.50	*Stipa aliena* Keng
P (%)	0.65	
Aboveground biomass (g/m^2^)	343.52	*Potentilla bifurca* Linn.
Grass height (cm)	16.12	

Note: Crude protein (CP); crude fat (EE); dry matter (DM); crude ash (Ash); neutral detergent fiber (NDF); acid detergent fiber (ADF); hemicellulose (HCEL).

**Table 2 foods-13-00679-t002:** Primer Information.

Items	(5′–3′)	Tm/°C	Length/bp	Gene Sequence No.
*β-actin*	F: AGCCTTCCTTCCTGGGCATGGA R: GGACAGCACCGTGTTGGCGTAGA	60	113	NM_001009784.3
*LPL*	F: CCTGGAGTGACGGAATCTGTG R: CCACGATGACGTTGGAGTCT	60	160	NM_001009394.1
*SCD*	F: TCACATTGATCCCCACCTGC R: CCGAGCTTTGTAGGTTCGGT	60	125	NM_001009254.1
*FASN*	F: CTTAACAGCACGTCCCCCAT R: TCCTCGGGCTTGTCTTGTTC	60	149	XM_027974304.2
*PPARγ*	F: CTTGCTGTGGGGATGTCTCA R: TTCAGTTGGTCGATGTCGCT	60	104	NM_001100921.1
*FABP4*	F: AGAAGTGGGTGTGGGCTTTG R: CTGGCCCAATTTGAAGGACATC	60	142	NM_001114667.1

**Table 3 foods-13-00679-t003:** Composition and content of SFAs in three muscle tissues of Tibetan sheep of different ages (%).

SFA	Tissue	4M	1.5Y	3.5Y	6Y
Butyric acid C4:0	Ldm	0.43 ± 0.326 ab	0.57 ± 0.259 ab	0.45 ± 0.106 b	0.84 ± 0.247 a
	Fm	0.64 ± 0.117 a	0.41 ± 0.095 b	0.78 ± 0.023 a	0.21 ± 0.020 c
	Hm	0.26 ± 0.039 c	0.73 ± 0.051 a	0.36 ± 0.033 b	0.32 ± 0.008 b
Tridecanoic acid C13:0	Ldm	0.85 ± 0.291 a	1.28 ± 0.229 a	0.63 ± 0.025 a	0.86 ± 0.089 a
	Fm	0.94 ± 0.004 b	0.91 ± 0.108 b	1.30 ± 0.065 a	0.56 ± 0.023 b
	Hm	0.71 ± 0.101 c	1.40 ± 0.160 a	0.94 ± 0.245 b	0.73 ± 0.078 c
Myristic acid C14:0	Ldm	4.83 ± 0.854 a	1.80 ± 0.237 b	2.14 ± 0.040 a	1.78 ± 0.012 b
	Fm	2.47 ± 0.180 a	2.14 ± 0.111 a	1.68 ± 0.203 b	2.02 ± 0.004 a
	Hm	4.89 ± 0.552 a	1.94 ± 0.240 a	1.97 ± 0.345 a	2.00 ± 0.069 a
Pentadecanoic acid C15:0	Ldm	0.51 ± 0.132 a	0.68 ± 0.132 a	0.25 ± 0.049 a	0.52 ± 0.090 a
	Fm	0.38 ± 0.036 b	0.39 ± 0.107 b	0.66 ± 0.007 a	0.42 ± 0.002 b
	Hm	0.59 ± 0.023 a	0.40 ± 0.016 a	0.36 ± 0.095 a	0.47 ± 0.034 a
Daturic acid C17:0	Ldm	0.84 ± 0.013 b	1.10 ± 0.167 a	1.19 ± 0.090 a	0.86 ± 0.127 b
	Fm	1.02 ± 0.514 a	1.23 ± 0.035 a	0.94 ± 0.134 b	1.11 ± 0.012 a
	Hm	0.76 ± 0.126 c	0.93 ± 0.010 b	0.98 ± 0.083 b	1.11 ± 0.016 a
Stearic acid C18:0	Ldm	15.89 ± 0.065 c	20.68 ± 1.924 b	24.35 ± 0.902 a	17.06 ± 0.199 c
	Fm	18.99 ± 0.277 c	21.73 ± 0.871 b	19.32 ± 0.811 c	23.23 ± 0.380 a
	Hm	17.59 ± 0.539 b	17.19 ± 0.673 b	21.59 ± 0.555 a	21.02 ± 0.096 a
Docosa-octadecanoic acid C22:0	Ldm	0.43 ± 0.083 ab	0.63 ± 0.261 a	0.34 ± 0.029 a	0.52 ± 0.045 a
	Fm	0.75 ± 0.123 b	0.66 ± 0.039 b	0.88 ± 0.004 a	0.41 ± 0.035 c
	Hm	0.35 ± 0.036 c	0.81 ± 0.209 a	0.59 ± 0.006 b	0.47 ± 0.095 b

Note: Different lowercase letters in the same column indicate significant differences between different altitudes at *p* < 0.05 level. 4M: 4-month age group; 1.5Y: 1.5 years age group; 3.5Y: 3.5 years age group; 6Y: 6 years age group; Ldm: Longissi-mus dorsi; Fm: Foreleg muscle; Hm: Hindleg muscle.

**Table 4 foods-13-00679-t004:** Composition and content of UFAs in three muscle tissues of Tibetan sheep of different ages (%).

UFA	Tissue	4M	1.5Y	3.5Y	6Y
Myristoleic acid C14:1	Ldm	0.66 ± 0.177 a	0.87 ± 0.305 a	0.58 ± 0.022 a	0.67 ± 0.029 a
	Fm	0.66 ± 0.178 a	0.78 ± 0.210 b	1.19 ± 0.056 a	0.39 ± 0.027 c
	Hm	0.43 ± 0.011 b	1.04 ± 0.222 a	0.62 ± 0.070 b	0.64 ± 0.074 b
Palmitoleic acid C16:1	Ldm	0.98 ± 0.315 b	1.19 ± 0.019 b	1.08 ± 0.058 c	1.50 ± 0.050 a
	Fm	1.31 ± 0.107 a	1.13 ± 0.035 a	1.26 ± 0.156 ab	1.06 ± 0.047 b
	Hm	1.34 ± 0.023 a	1.24 ± 0.057 a	1.31 ± 0.216 a	1.25 ± 0.029 a
Cis-10-heptadecenoic acid C17:1	Ldm	0.37 ± 0.011 d	0.67 ± 0.004 a	0.50 ± 0.012 c	0.61 ± 0.012 b
	Fm	0.49 ± 0.040 b	0.60 ± 0.070 a	0.51 ± 0.023 a	0.58 ± 0.030 a
	Hm	0.41 ± 0.021 b	0.63 ± 0.040 a	0.58 ± 0.011 a	0.59 ± 0.037 a
Elaidic acid C18:1n9t	Ldm	3.23 ± 0.139 b	2.54 ± 0.291 b	4.83 ± 0.165 a	2.58 ± 0.339 b
	Fm	3.74 ± 1.042 ab	3.31 ± 0.476 b	2.65 ± 0.953 b	4.80 ± 0.015 a
	Hm	2.39 ± 0.148 b	2.86 ± 0.617 b	3.24 ± 0.985 ab	4.50 ± 0.405 a
Oleic acid C18:1n9c	Ldm	30.57 ± 1.638 c	34.59 ± 1.257 ab	32.70 ± 1.434 b	36.62 ± 1.026 a
	Fm	30.00 ± 0.110 c	34.86 ± 0.989 a	31.37 ± 0.844 a	32.45 ± 0.380 a
	Hm	28.43 ± 0.626 b	30.59 ± 1.993 a	32.50 ± 1.197 a	32.65 ± 0.833 a
Linoleic acid C18:2n6c	Ldm	5.49 ± 1.944 a	6.46 ± 0.972 a	4.53 ± 0.315 a	4.89 ± 0.483 a
	Fm	6.11 ± 0.069 b	4.81 ± 1.261 b	8.33 ± 0.292 a	4.39 ± 0.033 b
	Hm	5.80 ± 0.690 b	8.77 ± 1.093 a	6.86 ± 0.367 b	5.88 ± 0.345 b
Alpha linoleic acid C18:3n3	Ldm	0.87 ± 0.157 ab	0.61 ± 0.011 a	0.80 ± 0.095 a	1.18 ± 0.493 a
	Fm	0.59 ± 0.189 b	0.69 ± 0.073 b	0.59 ± 0.191 b	1.56 ± 0.093 a
	Hm	2.17 ± 0.071 b	3.29 ± 0.612 a	2.12 ± 0.082 b	1.66 ± 0.731 b
Arachidonic acid C20:4n6	Ldm	2.32 ± 0.904 a	2.77 ± 0.452 a	1.58 ± 0.248 a	2.15 ± 0.062 a
	Fm	3.85 ± 0.498 a	1.88 ± 0.466 b	3.59 ± 0.200 a	1.59 ± 0.094 b
	Hm	3.33 ± 0.530 ab	4.17 ± 0.013 a	2.53 ± 0.730 b	1.97 ± 0.116 b
Cis-5,8,11,14,17-eicosapentaen-oic acid C20:5n3	Ldm	0.55 ± 0.180 b	1.49 ± 0.253 a	0.56 ± 0.138 b	1.38 ± 0.185 a
	Fm	1.91 ± 0.255 a	0.90 ± 0.169 b	1.55 ± 0.335 a	0.77 ± 0.047 b
	Hm	2.09 ± 0.073 a	2.09 ± 0.073 a	1.10 ± 0.209 b	0.98 ± 0.149 b
Docosahexaenoic acid C22:6n3	Ldm	0.86 ± 0.296 a	1.38 ± 0.221 a	0.73 ± 0.117 a	1.23 ± 0.126 a
	Fm	0.47 ± 0.246 a	0.47 ± 0.246 a	0.33 ± 0.005 a	0.21 ± 0.008 a
	Hm	0.19 ± 0.054 b	0.33 ± 0.029 a	0.24 ± 0.010 b	0.22 ± 0.019 b

Note: Different lowercase letters in the same column indicate significant differences between different altitudes at *p* < 0.05 level. 4M: 4-month age group; 1.5Y: 1.5 year age group; 3.5Y: 3.5 years age group; 6Y: 6 years age group; Ldm: Longissi-mus dorsi; Fm: Foreleg muscle; Hm: Hindleg muscle.

**Table 5 foods-13-00679-t005:** Composition and content of total fatty acids in three muscle tissues of Tibetan sheep of different ages.

Fatty Acid	Tissue	4M	1.5Y	3.5Y	6Y
UFA/%	Ldm	47.58 ± 2.322 b	52.56 ± 1.669 b	47.87 ± 0.573 b	52.80 ± 1.604 a
	Fm	49.31 ± 0.385 b	49.42 ± 2.001 b	51.38 ± 0.672 a	47.80 ± 0.401 b
	Hm	47.24 ± 0.728 c	55.00 ± 0.034 a	51.08 ± 1.777 b	50.32 ± 0.784 b
SFA/%	Ldm	23.78 ± 0.074 c	26.75 ± 1.657 b	29.37 ± 0.916 a	22.46 ± 0.282 c
	Fm	25.20 ± 0.692 b	27.49 ± 0.509 a	25.58 ± 0.827 b	27.97 ± 0.465 a
	Hm	25.16 ± 0.014 c	23.41 ± 0.747 b	26.80 ± 0.451 a	26.12 ± 0.451 a
MUFA/%	Ldm	35.80 ± 1.649 c	39.85 ± 1.228 b	39.68 ± 1.294 b	41.98 ± 0.970 a
	Fm	36.31 ± 0.673 b	40.67 ± 0.861 a	36.97 ± 0.032 b	39.28 ± 0.409 a
	Hm	33.00 ± 0.444 c	36.36 ± 1.582 b	38.24 ± 1.061 a	39.62 ± 0.727 a
PUFA/%	Ldm	11.78 ± 3.970 a	12.71 ± 1.893 a	8.19 ± 0.721 a	10.82 ± 1.181 a
	Fm	13.00 ± 0.288 b	8.75 ± 1.722 b	14.41 ± 0.654 a	8.52 ± 0.009 b
	Hm	14.25 ± 0.284 b	18.64 ± 1.615 a	12.84 ± 1.214 b	10.71 ± 0.401 b
P/S	Ldm	0.50 ± 0.169 a	0.41 ± 0.161 ab	0.28 ± 0.015 b	0.50 ± 0.069 a
	Fm	0.52 ± 0.026 b	0.32 ± 0.067 b	0.59 ± 0.007 a	0.31 ± 0.006 b
	Hm	0.57 ± 0.011 b	0.80 ± 0.095 a	0.48 ± 0.038 b	0.48 ± 0.039 b

Note: Different lowercase letters in the same column indicate significant differences between different altitudes at *p* < 0.05 level. 4M: 4-month age group; 1.5Y: 1.5 year age group; 3.5Y: 3.5 years age group; 6Y: 6 years age group; Ldm: Longissi-mus dorsi; Fm: Foreleg muscle; Hm: Hindleg muscle.

## Data Availability

The microbial sequence data are available in the NCBI database under accession PRJNA1063531.

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
