# Peer review of "Study of the Interactions between Muscle Fatty Acid Composition, Meat Quality-Related Genes and the Ileum Microbiota in Tibetan Sheep at Different Ages"

_foods, 2024, doi:10.3390/foods13050679_

Round 1

Reviewer 1 Report

Comments and Suggestions for Authors

The title of the manuscript is bit complicated, Authors could try to elaborate it.

line 21 what Authors mean by "food value"?

"different ages" in the list of keywords is not very specific, "meat" & "gene expression" should be added

line 29 largest - mean how many heads?

lines 38-42 UFAs -multiple fatty acids. And please add information about the risks related with fat rancidity

what about fat composition and rumen biota?

Please check : https://www.nature.com/articles/s41598-022-08761-5

https://doi.org/10.3389/fnut.2021.701511

https://doi.org/10.1186/s12917-020-02477-2

https://doi.org/10.3389/fmicb.2022.1047744

M&M the information about animals from experimental groups is very limited and must be improved.

line 91 longissimus dorsi & what exatcle muscles from legs?

RNA extraction - the size of the sample?

lines 152& 214& 238& 260&299 and further - please check sentence in bold

Table 2 needs re-organisation as in current version is hard to analyse it

Fig. 10 is a valuable addition to the manuscript

Discussion: different oils from plant or animal origin are more typically adviced as a sources of 'good' FA in the diet - how the Authors would discuss the changes in fats during heat treatment of the meat. esp. frying?

line 648 please avoid comparisons of ruminants with humans

line 660 Authors did not conducted any tests related to meat quality/sensory analysis, so this conclusion is not based on the results

Comments on the Quality of English Language

the language needs corrections

please check e.g. gut microbiota is.. not 'are'

Author Response

The title of the manuscript is bit complicated, Authors could try to elaborate it.

Reply: Thanks to the reviewers for their questions. We have changed the title of the article to: “Study of the interactions between muscle fatty acid composition, meat quality-related genes and the ileum microbiota in Tibetan sheep at different ages”.

Line 21 what Authors mean by "food value"?

Reply: Thank you to the reviewers for their suggestions. I'm sorry we didn't phrase it very well. What we mean is “edible value”, and we have made further changes to the abstract of the article.

"Different ages" in the list of keywords is not very specific, "meat" & "gene expression" should be added.

Reply: Thank the reviewer for asking this question. We have added keywords according to the reviewer's suggestion.

Line 29 largest - mean how many heads?

Reply: Thanks to the reviewer for asking this question. After checking the latest data report, we found that the number of Tibetan sheep is 25 million. It has been supplemented in the article.

Lines 38-42 UFAs -multiple fatty acids. And please add information about the risks related with fat rancidity.

Reply: Thank the reviewer for asking this question. We have added the description of fatty acid rancidity to the description of polyunsaturated fatty acids.

What about fat composition and rumen biota? Please check:

https://www.nature.com/articles/s41598-022-08761-5

https://doi.org/10.3389/fnut.2021.701511

https://doi.org/10.1186/s12917-020-02477-2

https://doi.org/10.3389/fmicb.2022.1047744

Reply: Thank the reviewer for asking this question. We have supplemented the content of the article based on your recommended references.

M&M the information about animals from experimental groups is very limited and must be improved.

Reply: Thank you for your advice. We supplement the detailed data of experimental animals in lines 98-103 of the paper.

Line 91 longissimus dorsi & what exatcle muscles from legs?

Reply: Thank you for your question. We added precisely in line 112 of the article that the Longissimus dorsi is the muscle between the second thoracic vertebra and the second thoracic vertebra, the front leg is the triceps, and the back leg is the biceps femoris.

RNA extraction - the size of the sample?

Reply: Thank you for asking this question. Due to our negligence, we did not give a detailed description. Total RNA was extracted from longissimus dorsalis muscle and leg muscle tissues of Tibetan sheep using TRIzol kit (Invitrogen, Carlsbad, CA, USA), and specific experimental procedures were performed according to the kit guidelines. About 100 mg was sampled during RNA extraction.

Lines 152& 214& 238& 260&299 and further - please check sentence in bold.

Reply: Thank you for asking this question. We have made changes in the article.

Table 2 needs re-organisation as in current version is hard to analyse it.

Reply: Thank you for asking this question. We have reformatted Table 2.

Fig. 10 is a valuable addition to the manuscript.

Reply: Thank you very much for appreciating our work and for your thorough reading of our manuscript and your comments, which we believe contribute a lot to the improvement of the paper.

Discussion: different oils from plant or animal origin are more typically adviced as a source of ' good' FA in the diet - how the Authors would discuss the changes in fats during heat treatment of the meat. esp. frying?

Reply: Thank you for your question. We discuss in detail in lines 562-572 the effects of heat treatment on fatty acids and how to avoid the decline in fatty acid nutrient levels in muscle caused by high temperature cooking.

Line 648 please avoid comparisons of ruminants with humans

Reply: Thanks to the reviewer for asking this question. We have made changes in the article.

line 660 Authors did not conducted any tests related to meat quality/sensory analysis, so this conclusion is not based on the results.

Reply: Thank you in particular for your suggestions, which we have redescribed in lines 660-671 of the article.

Reviewer 2 Report

Comments and Suggestions for Authors

The authors attemp to answer how the fatty acid composition is regulated by genes in an ambitious way. The manuscript is overloaded with information but there is no "message to take home".

The title must be corrected. It does not make sense.

Test animals.

How many animals did you have from each age group? From how many difference flocks did you select them?

You cannot have animals from different age group that also come from different herds and subsequently different diets. These parameters should be taken into account.

Diet composition should be reported.

The method you used for fatty acid analysis is the O'Fallon method. The correct method after hexane addition and filtration would be to evaporate to dryness and then reconstitute in 1 ml of hexane. It is not clear why you partially evaporated the sample. (Lines 109-122).

Fatty acid composition

Please check that the names of individual fatty acid are correct. For example C14:1 is myristoleic and not myristic

It is more logical to present first the information of individual fatty acids (Tables 3 and 4) and then the information on lipid classes (Table 2).

Conclusion

What should be done in order to avoid having meat with an unfavourabe fatty acid composition. What is the best slaughter age?

There are too many figures. The numbering is not correct and there are figures that the title is missing in some of them.

Data presentation in the Tables. You cannot have 3 decimal points in the mean value and 3 decimal points in SD. The SD should have one more decinal point than the mean value i.e. two decimal points in the mean values and 3 decimal points in the SD values.

Comments on the Quality of English Language

Extensive English language corrections are required.

Author Response

The authors attemp to answer how the fatty acid composition is regulated by genes in an ambitious way. The manuscript is overloaded with information but there is no "message to take home".

Reply: Thank you for your comments, we have revised the whole text.

The title must be corrected. It does not make sense.

Reply: Thanks to the reviewers for their questions. We have changed the title of the article to: “Study of the interactions between muscle fatty acid composition, meat quality-related genes and the ileum microbiota in Tibetan sheep at different ages”.

Test animals:

1. How many animals did you have from each age group? From how many difference flocks did you select them?

Reply: Thanks to the reviewer for raising this question. In order to ensure the accuracy of the experiment, we randomly selected 6 Tibetan sheep of 4 ages from the same herd (about 300 sheep).

2. You cannot have animals from different age group that also come from different herds and subsequently different diets. These parameters should be taken into account.

Reply: Thank the reviewer for asking this question. In order to reduce experimental error, our test animals are all from the same flock and are under traditional natural grazing management, without any supplementary feeding. The sampling time was August, and the nutrient level of forage was mentioned in line 103 of the article.

Diet composition should be reported.

Reply: Thank the reviewer for asking this question. We have added detailed dietary components in line 118 of the article.

The method you used for fatty acid analysis is the O'Fallon method. The correct method after hexane addition and filtration would be to evaporate to dryness and then reconstitute in 1 ml of hexane. It is not clear why you partially evaporated the sample. (Lines 109-122).

Reply: Thank the reviewer for asking this question. We apologize for the ambiguity in our statement. The correct extraction method has been described in lines 122-136 of the article.

Fatty acid composition:

1. Please check that the names of individual fatty acid are correct. For example C14:1 is myristoleic and not myristic.

Reply: Thank the reviewer for asking this question. We have modified C14:1 in line 262 and line 269 of the paper, and have checked and modified the writing of other fatty acids in the paper in detail.

2. It is more logical to present first the information of individual fatty acids (Tables 3 and 4) and then the information on lipid classes (Table 2).

Reply: Thanks to the reviewer for asking this question. We have changed the positions of Table 2 and Table 3 and 4 to make the paper more compact.

Conclusion:

1. What should be done in order to avoid having meat with an unfavourabe fatty acid composition. What is the best slaughter age?

Reply: Thank you for your advice. The best slaughter age of Tibetan sheep is 1.5 years old, so we can choose to slaughter at 1.5 years old, and the content of beneficial fatty acids in longisone muscle is higher than that in leg muscle, so try to avoid eating less meat containing a lot of unfavorable fatty acids. We have added content in lines 660-671 of the article.

2. There are too many figures. The numbering is not correct and there are figures that the title is missing in some of them.

Reply: Thank you for asking this question. Due to the occurrence of redundant figures in our own question, we have reviewed the full text in detail and made revisions.

3. Data presentation in the Tables. You cannot have 3 decimal points in the mean value and 3 decimal points in SD. The SD should have one more decinal point than the mean value i.e. two decimal points in the mean values and 3 decimal points in the SD values.

Reply: Thank you for your question. Due to our own problems resulting in incorrect data processing, we have conducted a detailed review of the full text and made changes. See Table 3, Table 4, Table 5.

Round 2

Reviewer 1 Report

Comments and Suggestions for Authors

Thank you for careful following my previous suggestions.

I see serious improvment of the manuscript.

Reviewer 2 Report

Comments and Suggestions for Authors

-

Comments on the Quality of English Language

-